# Fabrication of a Novel (PVDF/MWCNT/Polypyrrole) Antifouling High Flux Ultrafiltration Membrane for Crude Oil Wastewater Treatment

**DOI:** 10.3390/membranes12080751

**Published:** 2022-07-30

**Authors:** Banan Hudaib, Rund Abu-Zurayk, Haneen Waleed, Abed Alqader Ibrahim

**Affiliations:** 1Chemical Engineering Department, Faculty of Engineering Technology, Al-Balqa Applied University, Amman 11134, Jordan; 2Hamdi Mango Center for Scientific Research, The University of Jordan, Amman 11942, Jordan; r.abuzurayk@ju.edu.jo (R.A.-Z.); alkader.aa@yahoo.com (A.A.I.); 3Nanotechnology Center, The University of Jordan, Amman 11942, Jordan; 4Department of Chemical Engineering, Jordan University of Science and Technology, Irbid 22110, Jordan; enghaneenwaleed@yahoo.com

**Keywords:** ultrafiltration, membrane, PVDF, polypyrrole, crude oil, wastewater

## Abstract

The present work deals with the fabrication of novel poly(vinylidene fluoride) (PVDF)/Multi-wall Carbon Nanotubes (MWCNT)/Polypyrrole (PPy) ultrafiltration membrane by phase inversion technique for the removal of crude oil from refinery wastewater. In situ polymerization of pyrrole with different concentrations of MWCNT ranging from 0.025 wt.% to 0.3 wt.% in PVDF prepared solutions. Measurement of permeability, porosity, contact angle, tensile strength, zeta potential, rejection studies and morphological characterization by scanning electron microscopy (SEM) were conducted. The results showed that membrane with (0.05% MWCNT) concentration had the highest permeability flux (850 LMH/bar), about 17 folds improvement of permeability compared to pristine PVDF membrane. Moreover, membrane rejection of crude oil reached about 99.9%. The excellent performance of this nanocomposite membrane suggests that novel PVDF modification with polypyrrole had a considerable effect on permeability with high potential for use in the treatment of oily wastewater in the refinery industry.

## 1. Introduction

A credible energy supply influences the new lifestyle. Fossil fuel has been considered the most compatible energy source among the various energy sources for centuries; it is even considered the fundamental energy source [1]. The petroleum processing industry produces a considerable amount of water at different stages, e.g., thermal cracking, distillation, and catalytic cracking, producing a large amount of wastewater.Petroleum industry wastewater contains a variety of organic and inorganic pollutants, such as sulfides, phenol, heavy metals, hydrocarbons, etc. [2,3]. The amount of wastewater produced from the petroleum refining process is estimated to be about 1.6 times that of processed crude oil [4]. Crude oil wastewater contains various percentages of hydrocarbons, phenols, and dissolved minerals [5] depending on the origin and storage conditions, affecting crude oil sludge composition. Generally, it comprises about 10–30 wt.% hydrocarbons, 5–20 wt.% solids and 50–85 wt.% water [6,7]. In crude oil wastewater, the hydrocarbons (benzene, toluene, ethylbenzene, and xylenes), Phenols, and dissolved minerals are carcinogenic and toxic to human and aquatic life [8,9].

In Jordan Petroleum Refinery, different oily fluids are collected in substantial waste pools with estimated flow rates of more than 15 m^3^/day. Considering that Jordan is an arid/semi-arid country, treating these large wastewater quantities using emerging clean technologies like membranes is mandatory to recycle and reuse produced water and avoid its harmful effects on the environment and human health.

Several membrane technologies, mainly pressure-driven membrane separation processes, have been applied for petroleum wastewater treatment, including microfiltration (MF) [10], ultrafiltration (UF) [11,12], and nanofiltration (NF) [13]. 

A perfectly designed membrane with excellent permeability and porosity will inherently improve flux and better economics [14,15]. Membranes pore size and materials will depend on the application for which it would be used [14].

In contrast to other membrane processes, ultrafiltration (UF) has a wide range of uses, including protein filtration, bacterial elimination, food product fractionation, oil-water separation, and others [16]. Specialized polymers like poly(vinylidene fluoride) (PVDF), poly(ether sulfone) (PES), polyacrylonitrile (PAN), and polysulfone (PS) are the cornerstone materials in the UF membrane fabrication owing to their good performances and unique properties such as chemical stability, high efficiency, distinguished mechanical properties and heat resistance [17,18]. However, the drawbacks of UF membranes fabricated from these materials include poor surface wettability, which results in severe membrane fouling resulting from solute–hydrophobic membrane interactions [19]. 

Crozes et al. [20], who studied the effect of membrane hydrophilicity and organic compound polarity on membrane fouling, concluded that hydrophobic polymers appeared to be the fundamental foulant material. Therefore, membrane surface modification using hydrophilic additives employing physical adsorption, polymer blend [21], and plasma modification [22] were carried out. The hydrophilic surface modifications aim to fabricate high flux membranes with antifouling properties. Unfortunately, surface modifications might adversely affect the membrane structure, causing weak mechanical properties and inclined rejection [23]. Asatekin and Mayes [11] reported using ultrafiltration (UF) membranes for refinery wastewater treatment, and results showed ultimately low chemical oxygen demand (COD) removal rates, between 41 and 44%, attributed to high dissolved organics contents. However, ongoing research improves UF membrane performance by incorporating nanomaterials in the membrane fabrication process [23]. 

Incorporating nanomaterials, which can exhibit favorable properties that differ from bare polymers, is a promising new trend in membrane fabrication technology [24]. Nanocomposite membranes are evolved by using engineered nanoparticles into porous membranes [25] or blending them with polymeric membranes [26]. Membranes were improved using silica, graphite, zeolite, metal oxide nanoparticles, or carbon nanotubes to enhance the membrane flux and antifouling properties [27,28,29]. Among various nanoparticles, carbon nanotubes (CNTs) have attracted considerable attention since their discovery by Iijima in 1991. CNTs are tiny graphite cylinders closed at the end by half C60 with single or multiple walls (SWCNT or MWCNT) [30]. CNTs have a diameter of less than 100 nm. They have distinguished characteristics, such as distinctive mechanical properties (flexibility and stiffness) and high electrical and thermal conductivities [31], with remarkable water treatment efficiency in removal of various chemical and biological contaminants [31,32]. The CNTs dispersion capacity in a variety of polymer matrices such as PVDF, PS, PES, PAN, Polyamide, and Cellulose Acetate (CA) has attracted a lot of attention [33]. When used as an adsorbent media, CNT can remove a wide spectrum of contaminants, including heavy metals, metalloids, and organics [34,35,36,37]. The excellent adsorbent properties are assigned to the increased specific surface area [37], the mesoporous (super-nanoporous) structure and the fewer CNTs surface negative charge in addition to the effective carbon nanotube and aromatic compounds π-π stacking interaction [38]. These outstanding properties made them a convenient choice for polymer composites modification. 

Conducting polymers have different applications, from passive coatings to effective materials with functional electronic, energy storage, and mechanical properties [39]. They can be compiled using oxidation or reduction reaction (doping process) [40]. Among different conducting polymers, polypyrrole (PPy) has attracted researchers’ attention as an adsorbent for its outstanding properties such as high conductivity, non-toxicity, easy preparation methods, electrical and thermal stability, strong mechanical properties, availability at an industrial scale, and remarkable capacity for heavy metals adsorption from water effluents [41,42,43,44]. Thus, these remarkable properties granted polypyrrole many prospective applications in electronic and electrochromic devices, supercapacitors, corrosion protection, and fabrication of membranes with good separation and antifouling properties [45,46,47].

This work fabricated a novel (MWCNT/PPy/PVDF) ultrafiltration membrane to treat crude oil wastewater. A simple in-situ polymerization method was used to fabricate polypyrrole-coated MWCNT using Ammonium peroxydisulfate (APS) as an oxidant. The modified membranes showed enhanced water permeability and antifouling properties.

## 2. Experimental Work (Materials and Methods)

### 2.1. Materials

PVDF (Kynar 460) from Arkema has a broad molecular weight distribution with density of 1760 kg/m^3^ (ISO 1183, and melt volume flow rate of 5.6 cm^3^/10 min (ISO1133). Multi-walled Carbon nanotube (MWCNT, 98% carbon basis) from Aldrich (St. Louis, MO, USA), were used as received, Pyrrole (≥98%, FCC, FG) and Ammonium peroxydisulfate (APS) from Sigma Aldrich (St. Louis, MO, USA). N,N-Dimethylformamide (DMF) from Merck (Kenilworth, NJ, USA), Crude oil from Jordan Petroleum Refinery (Zarqa, Jordan).

### 2.2. Membrane Preparation

Synthesis of MWCNT/polypyrrole (PPy)

Membrane preparation started with the synthesis of MWCNT/PPy by in-situ oxidative polymerization of pyrrole on MWCNT; please note Figure 1. MWCNTs were dispersed by ultrasonication in DMF solvent using a probe sonicator (200 W) for 30 min, then pyrrole and APS were added to the mixture, and the solution was stirred for 48 h at room temperature.

b.Preparation of MWCNT/PPy/PVDF ultrafiltration membrane

Finally, 11.67 wt.% PVDF was added to the MWCNT/PPy composite solution and stirred for 24 h at 70 °C. The resultant casting solution was degassed by ultrasonication for one hour, then cast on a glass plate using a knife blade applicator with 250 µm thickness. Then, the casted film was immersed in a water bath to form a porous membrane at room temperature. Table 1 shows the different compositions of modified fabricated membrane casting solutions.

### 2.3. Emulsion Preparation

Crude oil (0.5 mL) and 0.5 mL of Polyoxyethylene-80 surfactant were diluted in 1 L distilled water to ensure homogeneity, and the mixture was stirred vigorously for 48 h. The resultant solution was tested using a UV-Vis spectrophotometer (Varian Cary 100, Varian medical systems, Crawley, UK), and the maximum wavelength was obtained at 358 nm. Moreover, the droplet size distribution was measured using ZetasizerXNem Tempus Malvern (Malvern, Uk), and the size distribution showed a maximum (96.1%) intensity at 400.1 nm; the measured oil properties are shown in Table 2.

### 2.4. Membrane Characterization

#### 2.4.1. Chemical Composition and Characterization of the Modified Membrane

Energy dispersive spectroscopy (EDS) was performed using A Phenom XL G2 scanning electron microscope (SEM, Thermo Fisher Scientific, Waltham, MA, USA) coupled with AXS EDS system. Furthermore, Fourier-transformation infrared spectroscopy (FTIR spectra, Model PerkinElmer, Waltham, MA, USA) was used to investigate the incorporation of PPy on MWCNT in modified membranes.

#### 2.4.2. Membrane Hydrophilicity (Contact Angle) and Zeta Potential

Membrane hydrophilicity was measured using (2.5–5) µL sessile droplets of deionized water by Theta Lite, Biolin Scientific (Stockholm, Sweden). (5–10) contact angle measurements were done at different points of the membrane sample and then averaged.

The surface charge of the membrane was measured using the potential streaming technique DLS (Zetatrac, Westborough, MA, USA). Experiments were conducted using deionized water at room temperature.

#### 2.4.3. Membrane Morphology and Structure

Images were taken for the fabricated membranes surfaces and cross-sections of the modified membranes using a Field Emission Scanning Electron Microscope (SEM, Quanta FEG 450, FEI company, Hillsboro, OR, USA). The samples were gold sputter-coated before the examination.

#### 2.4.4. Porosity and Average Pore Size Calculation

Membrane porosity was determined by using the gravimetric method. Membrane samples were initially immersed in water and then dried in an oven at 70 °C; Mass loss of the wet membrane after drying was measured. Membrane porosity (ε) was calculated by the equation below (1) [48]:(1)$$\varepsilon =\frac{\left(W_{W}-W_{d}\right)/\rho_{\mathrm{water}}}{\left(W_{W}-W_{d}\right)/\rho_{\mathrm{water}}+W_{d}/\rho_{p}}$$
where $W_{W}$: is the wet membrane weight in (g), $W_{d}\;:\mathrm{is}\;$the dry membrane weight (g), $\rho_{\mathrm{water}}$: is the pure water density which is equal (0.998 g·cm^−3^) and $\rho_{p}$: is the polymer density (as the inorganic content in the membrane matrix was small and $\rho_{p}$ was approximate to $\rho_{\mathrm{PVDF}}$ Which is equal to 1.765 g·cm^−3^). 

The mean pore size of the modified membranes was measured using the N_2_ adsorption-desorption isotherm (Brunauer-Emmett-Teller (BET)) method analyzer (Autosorb IQ, Quantachrome Instruments version 5.21, Boynton Beach, FL, USA) [49].

#### 2.4.5. Permeation Performance

Pure water flux and crude oil rejection were measured using high-pressure stirred cell (HP 4750, Sterlitech, Auburn, AL, USA) equipment. All experiments were carried out at room temperature and under the feed pressure of 0.2 MPa. The concentrations of crude oil in the permeation and feed solution were determined by UV- spectrophotometer (Varian Cary 100) at 358 nm. The flux and rejection were calculated using Formulaes (2) and (3), respectively:(2)$$J_{p}=\frac{V}{A\times ∆t}\;$$
(3)$$R=\left(1-\frac{C_{p}}{C_{F}}\right)\times 100\%$$
where *J_P_* was the membrane flux for pure water (L·m^−2^·h^−1^), *V* was the volume of permeate pure water (L), *A* was the membrane effective surface area (=14.6 cm^2^), and ∆t was the time of permeation (hr). *R* was the rejection of crude oil (%), and *C_P_* and *C_F_* were the concentrations of crude oil in the permeation and feed solution (mg/L), respectively.

#### 2.4.6. Antifouling Performance 

Membrane fouling is the material aggregation on the membrane surface or within its structure. Membrane fouling is one most important factors that determine membrane efficiency. Crude oil was used to assess the antifouling properties of the modified membranes. The antifouling properties of the fabricated membranes were studied as follows: initially, the permeation with pure water was done; followed by rejection with crude oil; and then the membrane was cleaned by shaking with (0.1 M) NaOH for one hour, then shaking with pure water for half-hour, after that permeation of pure water was done again. The flux recovery (*FR_w_*) and total fouling ratio (*R_t_*) were calculated using the following equations:(4)$$FR_{w}\left(\%\right)=\left(\frac{J_{p}{}_{w2}}{J_{p}{}_{w1}}\right)\times 100$$
(5)$$R_{t}\left(\%\right)=\left(1-\frac{H_{p}}{F_{w1}}\right)\times 100$$
where $J_{p}{}_{w1}$ = water permeability, $J_{p}{}_{w2}$ = water permeability of the cleaned membrane, $H_{p}$ = permeability of crude oil solution.

#### 2.4.7. Tensile Strength 

Membrane tensile strength and elongation-at-break were measured using Electromechanical Universal Tensile (BMT-E Series, BESMAK, Kazan, Turkey). The tensile rate was 5 mm/min. For each sample, three runs were done and then averaged.

## 3. Results and Discussion

### 3.1. EDS and FTIR Spectroscopy

The elemental analysis (EDS) diagram for the modified membrane is presented in Figure 2, and the results are summarized in Table 3. They show that carbon (56%), Fluorine (31%), and oxygen (8.8%) are the main composite chemical elements originating from the PVDF and MWCNT, with a low nitrogen percentage (4.2%) originating from PPy component.

The incorporation of MWCNT/PPy in the modified fabricated membranes was investigated using Fourier-transformation infrared spectroscopy (FTIR) of MWCNT and PPy/MWCNT complex samples, as explained in Figure 3

Figure 3 showed the FTIR spectra for pristine MWCNT and MWCNT/PPy complex mixture. The MWCNT displays broadband at 2990 cm^−1^, allocated to the OH stretching, and a band at 1078 cm^−1^, assigned to C=O stretching. [47]. While MWCNT/PPy spectrum exhibits a peak at 3338 cm^−1^, which is specified to the N-H hydrogen-bonded stretching vibration [50]. The sharp 1662 cm^−1^ peak is assigned to C-N asymmetrical ring stretching [51,52]. The peaks at 1506 cm^−1^ and 1439.7 cm^−1^ could be assigned to the symmetric and antisymmetric ring stretching modes of the PPy ring [47,53]. Otherwise, 1385 and 1257 cm^−1^ peaks could be assigned to C-N in-plane deformation. 1090 and 866 cm^−1^ Peaks can be allocated to pyrrole ring C-H in-plane and out-of-plane deformation, respectively [54].

The FTIR spectra suggested that the C-N band in the nanocomposites becomes more robust while the N-H, C-H, and C-C bonds become weaker [55,56]. This would support the formation of PPy and MWCNTs C-N bands and indicates that MWCNT/PPy incorporation in the nanocomposite was successful.

### 3.2. Membrane Hydrophilicity and Zeta Potential 

The surface wettability of the membrane is controlled by composition chemistry and the construction geometry of the membrane surface. The hydrophilic surface of the fabricated membranes can be explained by the hydrophilic nitrogen components (C-N) in the composition of the fabricated membranes (as can be seen from the FTIR spectra results), forming the final surface wettability of the membrane [47,53,57]. Moreover, the observed roughness structure of the produced membranes improves the surface’s wettability and reduces the contact angle (CA) [58]. As shown in Figure 4, The CAs of all PVDF/PPy/MWCNT (PPC) modified membranes are less than that of the non-modified membrane. The pristine PVDF CA was 85°, while MWCNT/PPy complex addition gave 80° for PPC-0.025 (0.025 wt.% MWCNT) membrane. It is noticed that as the MWCNT concentration increased in the PVDF matrix, the CA further decreased to the lowest value; thus, the highest hydrophilicity obtained was for PPC-0.3 (0.3 wt.% MWCNT) equal to 65°, suggesting that hydrophilicity enhancement is a result of modification with hydrophilic MWCNT/PPy complex in the fabricated membranes. 

This hydrophilicity improvement enhances water permeability, as discussed in the following sections.

Zeta potential is valuable for realizing the membrane surface and the oil contact interactions. Zeta potential substantially affects both membrane rejection ability and fouling behavior. In this experiment, the zeta potential was studied at pH 8.2. The zeta potential measurements Figure 5 showed that a pristine surface has a negative charge equal to −21.9 mV. When MWCNT alone was added to the PVDF membrane, the zeta potential increased to −16.1 mV. For PVDF/MWCNT/PPy modified membranes, a positively charged membrane surface equal to +15.8 mV was obtained. The PPy and MWCNT interaction are by a dopant effect, a distinctive chemical oxidation process converting polymers to conductive form [59]. 

In the chemical oxidation of the MWCNT/PPy nanocomposite during membrane fabrication, PPy-π electrons in the conjugated bond are removed, so benzenoid structure local relaxation into a quinoid form occurs and consequently creating a radical pair leading to the appearance of a positive charge [60,61,62]. Thus, the positively charged MWCNT/PPy transforms the negatively charged pristine PVDF membrane surface into a positively charged one. As presented in Figure 5, the zeta potential of the modified membrane’s surface became positive. Moreover, the crude oil charge was measured to be −15 mV; thus, the difference in charge between oil and modified membranes enhanced the rejection of oil, as shown in the rejection section.

### 3.3. Membrane Morphology and Structure

The SEM explored the morphological characteristic of the MWCNT/PPy nanocomposites. It was evident that PPy was grown and coating the MWCNTs, but at different thicknesses (Figure 6a,b). As the amounts of MWCNTs are less, thicker layers of PPy can be observed, while higher amounts of MWCNTs showed thinner layers of PPy. Thus, when the MWCNT percentage increased for PPC-0.3, the PPy layer thicknesses decreased, while for less MWCNT at PPC-0.05, the layer of PPy on MWCNT was increased; thus, more distribution of PPy amount existed on the surface of MWCNTs, similar results were reported by Baghdadi et al. [63].

Cross-sectional and membrane surface images of pristine, PP, PPC-0.05 and PPC-0.3 fabricated membranes are seen in Figure 7A–D. Results presented that the modified membranes displayed surface pore sizes as well as finger-like projections larger than pristine PVDF membranes, which is responsible for improved water permeability of the modified fabricated membranes; moreover, adding PPy increases surface roughness as presented in the membrane’s top surfaces because of the particles formed and aggregate on the membrane surface Figure 7A–D.

MWCNT/PPy/PVDF modified membrane’s cross-sectional structure showed large cavities and voids with asymmetrical finger-like projections. Furthermore, these membranes also formed smaller top layer pores, resulting in higher water flux and oil rejection [64].

The mechanism behind the formation of a highly porous layer with bigger pores in these membranes is likely related to the incorporation of the hydrophilic MWCNT/PPy complex, which in turn augmented the rate of diffusion between the solvent (DMF) and non-solvent (water) [64]. The width of the finger-like projections is related to an increase in the concentration of MWCNT/PPy, as can be seen in Figure 7C for PPC-0.05. On the other hand, with increasing concentration, the solution viscosity further increased, leading to smaller pore size and fewer finger-like projections [59] as a result of delayed diffusion rate. These results are consolidated by various research reported in the literature [48,59,65].

### 3.4. Porosity and Average Pore Size

The consequence of MWCNT/PPy matrix addition on the pore size and porosity for the modified membranes is shown in Table 4; increased pore sizes and porosity are noted in the modified membranes compared to pristine PVDF. Please note Table 4; MWCNT addition leads to improved porosity of the modified membranes by about 11% for PPC-0.025 (0.025 wt.% MWCNT) membrane compared to non-modified PVDF. Increasing MWCNT to 0.1% in PPC-0.1 gave the maximum porosity of 90%. The porosity increasing is due to nanocomposite inclusion in the modified membrane. 

MWCNT/PPy is a hydrophilic material that enhances the formation of a porous membrane structure by hastening the rate of diffusion between the solvent (DMF) and non-solvent (water), leading to an increase in porosity [59]. Furthermore, MWCNTs are materials with high surface area, thus forming macro-voids porous structures in the modified matrix [62]. Consequently, using MWCNT/PPy nanocomposite as a modifier for PVDF membrane could fabricate higher porosity membranes. On the contrary, the PPC-0.3 membrane showed a minute porosity decrease, which can be explained by the high MWCNT concentration (0.3 wt.%), leading to less MWCNT uniformity and hence increased mixture viscosity [48], which causes phase separation inhibition during membrane fabrication by delaying the demixing process, producing a low porous membrane [59,66].

The pore volume and pore size distribution for modified membranes analyzed by Barrett-Joyner-Halenda (BJH) method are presented in Table 5. The average pore size of modified membranes is (7.6 to 10.4) nm and is larger than pristine PVDF. The modified membranes pore sizes increased with MWCNT/PPy concentration increase. These results can be due to the addition of hydrophilic MWCNT/PPy, which enhances the diffusion rate between the solvent (DMF) and non-solvent (water), leading to the formation of a porous layer with larger pores. With increasing the MWCNT/PPy, the average pore size increased to reach a maximum for PPC-0.1 at 10.4 nm, compared with pristine PVDF at 3.05 nm. However, by increasing the concentration of MWCNT/PPy, the average pore size decreased slightly to 9.2 nm; this decrease in pore size could increase casting solution viscosity, consequently delaying the diffusion rate, leading to resultant smaller pore sizes.

### 3.5. Mechanical Properties of Modified Membranes

Tensile strength and elongation at the point of breakage of the modified PVDF/MWCNT/PPy membrane’s were tested. Results illustrated in Figure 8 indicated that modified membranes have excellent mechanical properties as the tensile strength of the modified membranes was higher than the pristine PVDF membrane. The tensile strength increment was related to the MWCNT concentration increment in the membrane composition, these results can be attributed to MWCNT rigidity and excellent mechanical properties, as well as suitability between MWCNT/PPy and the polymer matrix. Similar results were obtained in previous studies [48,63,67]. In contrast, in PPC-0.3 membrane showing the the highest MWCNT concentrations, the tensile strength decreased to 0.9 MPa, this could be related to the MWCNT aggregation due to high concentration, which accelerates membrane break; the same results were also shown by wang et al. [67]. Results of elongation at break for pristine and modified membranes showed mild reduction as the concentration of MWCNTs increased from 55% for pristine PVDF to 52% for PPC-0.3), possibly related to increased membranes rigidity and structural fragility due to MWCNT content as well as large cavities and pores formation [58]. Consequently, according to these results it can be concluded that MWCNT/PPy modification of the fabricated membranes comparatively enhances their mechanical strength.

### 3.6. Membranes Water Permeability and Rejection Efficiency 

The water flux of various membranes is illustrated in Figure 9 and Figure 10. The addition of MWCNT/PPy nanocomposite to the polymer matrix has significantly improved the performance of fabricated membranes. Please note Figure 9, which represents a comparison between 0.3 wt.% MWCNT/PVDF (PC), PVDF/PPy (PP) and 0.3 wt.% MWCNT/PPy/PVDF (PPC-0.3) and raw PVDF membranes. The lowest flux results with 30 LMH/bar are noted in non-modified raw PVDF due to its lower hydrophilicity and porosity as well as the smallest average pore size relative to the modified membrane. PP membrane has a water permeability up to 365 LMH/bar; 12 folds increase relative to raw non-modified PVDF membrane; this high flux can be explained by the effect of PPy oligomers and nanospheres migration in the fabrication process leading to increases in the pore sizes. Furthermore, the fabrication procedure method plays a role as the simple blending for PC membrane preparation (without PPy) as in 0.3% MWCNT leads to a low flux (80 LMH/bar) compared to the good result obtained using in situ polymerization method like in 0.3% MWCNT in PPC-0.3 (544.3 LMH/bar). This is likely because of poor dispersion and aggregation of MWCNT with PVDF in PC, which gives blocked pores and low porosity in the non-modified membrane. In opposition, PPC-0.3 proves to overcome aggregation with MWCNT/PPy homogenous dispersion (in-situ polymerization of pyrrole on the MWCNT surfaces), giving rise to the highest water flux results obtained from the four fabricated membranes presented below.

Figure 10 showed the permeability for different fabricated modified membranes (changing the presence of MWCNT). As seen in Figure 10, the addition of 0.025 wt.% MWCNT for the PPC-0.025 membrane resulted in a water flux increase up to 641 LMH/bar, while for PPC-0.05, MWCNT inclusion to 0.05 wt.%, leading to a maximum flux of 842 LMH/bar. Furthermore, PPC-0.1 showed a slight decrease to 726 LMH/bar for the last two modified membranes, while for PPC-0.3, permeability decreased to 544 LMH/bar.

These flux results are compatible with the results of porosity shown before. The larger the porosity, the less the flow resistance, enhancing the flux. Moreover, the increased hydrophilicity due to the inclusion of MWCNT/PPy nanocomposite improved the water permeability.

Moreover, MWCNT/PPy nanocomposite addition formed a porous well-developed finger-like top layer connected to the membrane’s lower layer. Furthermore, the addition of hydrophilic MWCNT/PPy increases the diffusion rate between the solvent (DMF) and non-solvent (water), enhancing the formation of a porous layer and increasing the hydrophilicity shown before. Thus, water molecules favorably adsorbed inside the membrane pore with minimum interactions facilitating more effortless flow through the membrane and increasing water flux as presented in the modified membranes [48,64,68]. Moreover, the results shown in Table 4 illustrated that the porosity and average pore size increased with MWCNT concentration increment, and hence water flux increased as stated in the Hagen Poiseulle equation [67]. On the other hand, with more MWCNT concentration addition like in 0.3 wt.% MWCNT for PPC-0.3 membranes, a denser top layer with smaller pores, was noted due to the casting solution’s high viscosity, reducing the solvent and non-solvent exchange [59]. The optimum concentration of added MWCNT that gave the highest membrane permeability results is 0.05 wt.% for PPC-0.05. Please see (Figure 10).

Crude oil rejection results in Table 6 showed a higher rejection efficiency of more than 99% for all modified membranes; Thus, PVDF/MWCNT/PPy modified membranes effectively separated water from crude oil. Photographs of the feed and permeate (Figure 11) showed clear filtrate compared to feed wastewater, with efficacy reaching 99.5% for all modified membranes.

The separation process of crude oil wastewater using a membrane is a sieving process that relies on size exclusion, membrane surface hydrophilicity, and surface charges (electrostatic interaction) [69]. For the sieving effect, larger-sized oil droplets are blocked by smaller membrane pores; as noticed previously, the average membrane pores size is between (7.6 to 10.4) nm, while the measured oil droplets size was about 400 nm.

The hydrophilic membrane characteristic is more effective for treating oily wastewater as the membrane allows water molecules to pass through and blocks oil droplets [70]. Moreover, the surface charge of the membrane is a vital factor in oil adsorption on the surface of the membrane due to the electrostatic interaction between the charged membrane surface and oil wastewater. As crude oil wastewater is negatively charged molecules (the measured zeta potential is -15 mV), an electrostatic interaction occurs between oil and the positively charged membrane surface, as shown previously in Figure 5 (+15.8 mV), causing oil molecules adsorption leading to high oil removal rate by the modified membranes [71].

A comparison between the results of this study and different studies in the literature that used PVDF membranes for oil wastewater removal is summarized in the following Table 7.

### 3.7. Fouling and Cleaning

During the crude oil wastewater filtration, oil droplets accumulated on the surfaces of the fabricated membranes, leading to a severe fouling and a severe drop in membrane flux [45]; as shown in Figure 12, crude oil significantly has low flux compared to pure water flux. Filtration test cycles were run for all modified membranes for about 210 min. Please see Figure 12. 

Results showed that using caustic soda as a chemical-cleaning method removed the oil layer and restored the flux to its original values for the produced modified membranes in contrast to pure PVDF, as shown in Figure 12. These results are attributed to the enhanced properties of MWCNT/PPy, which increased the membrane surface hydrophilicity and increased internal pores [46,76]. Moreover, the efficient cleaning method of the membrane surfaces uses caustic soda (NaOH). Caustic removes oil by hydrolysis and solubilization effect [47].

Filtration and rejection test cycles were conducted for PPC-0.05 membrane for about 300 min, as shown in Figure 13; oil rejection efficiency was measured every 60 min, and the membrane was washed with caustic soda every 60 min, and then the membrane was reused. Results showed a slight flux decline of the modified membrane with high oil rejection efficiency, almost reaching up to 99%, suggesting high stability of the modified membranes [48].

The suitable cleaning method and the flux recovery test for oil were conducted. Figure 14 showed flux recovery and the total fouling ratio results for the produced fabricated membranes.

As revealed in Figure 14, flux recovery (FRw) for the fabricated membranes after cleaning with caustic soda is between 88–93%, which is an excellent result as the crude oil caused considerable fouling. For total fouling ratio (Rt) was found to be between (80–89)%, as presented in Figure 14. This high recovery indicated the simple cleanability of the modified fabricated membranes compared to pure PVDF.

Although oil wastewater often causes sharp flux decline and significant fouling. It can be easily recovered using caustic soda.

## 4. Conclusions

A novel ultrafiltration (MWCNT/PPy/PVDF) membrane with enhanced water permeability, oil rejection, and antifouling properties was prepared by pyrrole in-situ polymerization. The addition of MWCNT/PPy to the PVDF polymer enhanced the membrane properties. Thus the new modified membranes showed higher hydrophilicity, porosity, and permeability. Hydrophilicity results showed a decrease in contact angle from 88 for pristine PVDF to 69 for PPC-0.1. PPC-0.05 shows super permeability with flux up to 850 LMH/bar; (about 20 times increase more than pristine PVDF). Furthermore, the Inclusion of MWCNT/PPy gave a high crude oil rejection, reaching 99.9% due to the electrostatic interaction and the sieving properties of MWCNT/PPy/PVDF modified membranes. The flux recovery increased from 50% for PVDF to 89.5% for PPC-0.1. MWCNT/PPy complex significantly enhanced the fabricated membranes’ separation and water permeability. In-situ polymerization technique leads to excellent dispersion of pyrrole and MWCNT in MWCNT/PPy complex. It is an excellent modifier for ultrafiltration membranes for future applications.

## Figures and Tables

**Figure 1 membranes-12-00751-f001:**
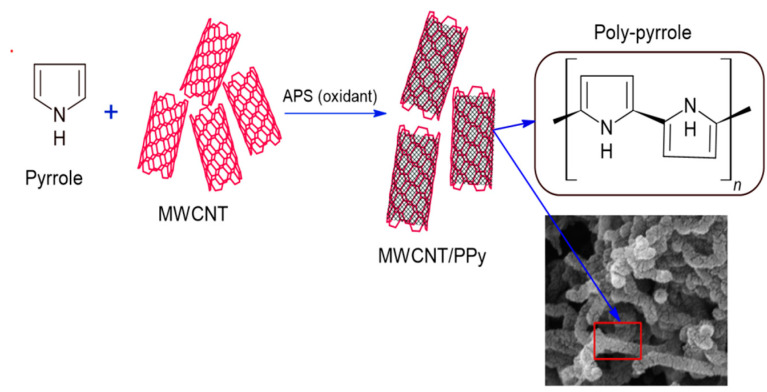
MWCNT/PPy synthesis method.

**Figure 2 membranes-12-00751-f002:**
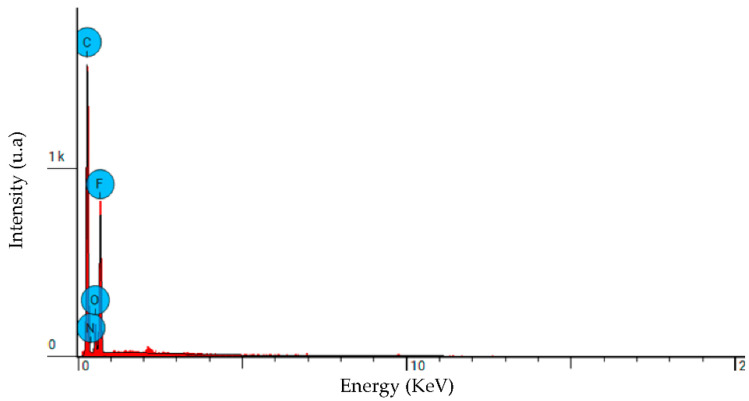
EDS spectrum of the PPC-0.05 modified membrane.

**Figure 3 membranes-12-00751-f003:**
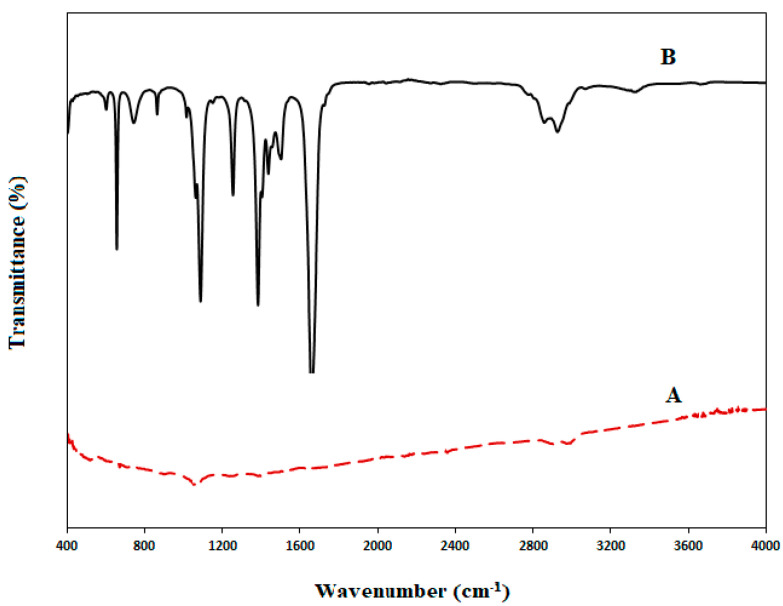
FTIR for A. MWCNT, B. MWCNT/PPy.

**Figure 4 membranes-12-00751-f004:**
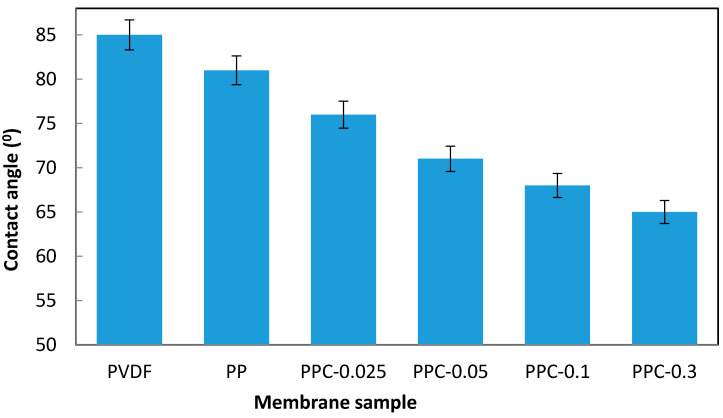
Hydrophilicity measurement for pristine and modified membranes.

**Figure 5 membranes-12-00751-f005:**
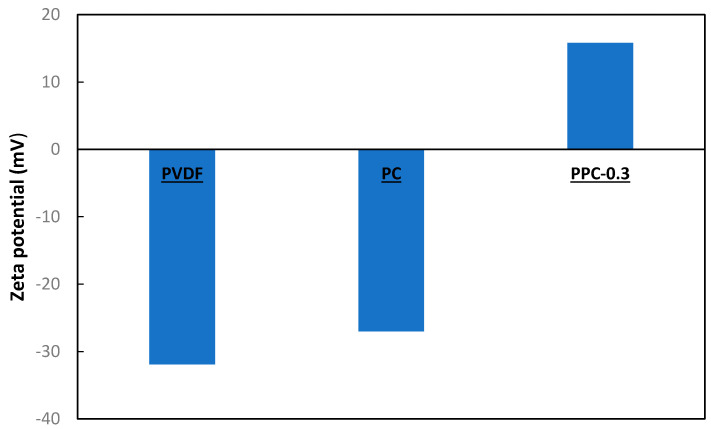
Zeta potential for modified membranes.

**Figure 6 membranes-12-00751-f006:**
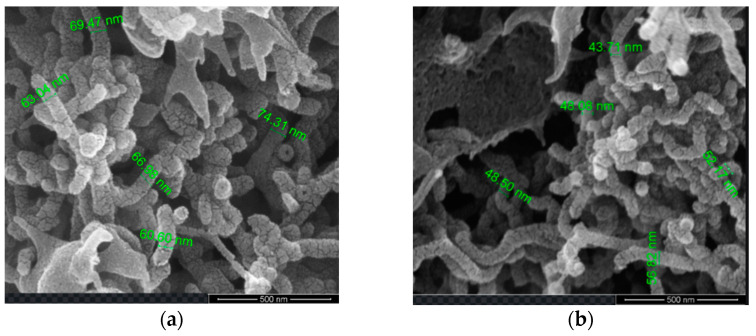
SEM images of MWCNTs/PPy nanocomposite: (**a**) PPC-0.3, (**b**) PPC-0.05.

**Figure 7 membranes-12-00751-f007:**
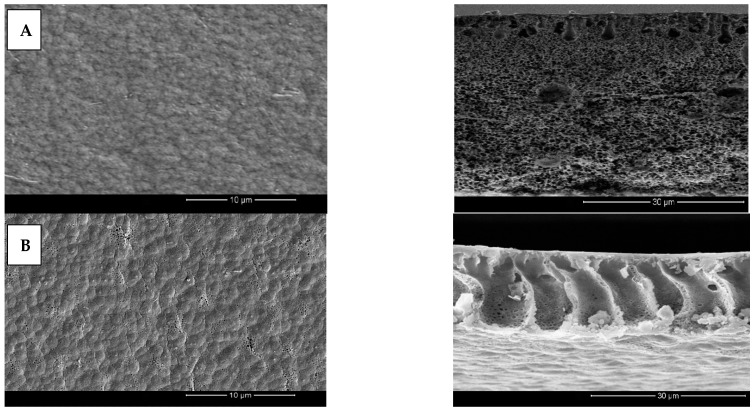
(**A**–**D**) Different modified Membrane’s top surface and cross-sectional SEM images at 10 µm and 30 µm magnifications respectively; (**A**) pristine, (**B**) PP, (**C**) PPC-0.05, (**D**) PPC-0.3.

**Figure 8 membranes-12-00751-f008:**
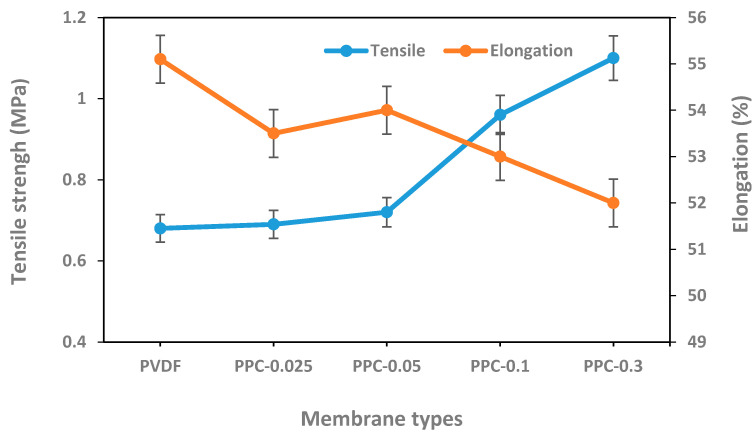
Tensile strength and elongation at break for pristine and modified membranes.

**Figure 9 membranes-12-00751-f009:**
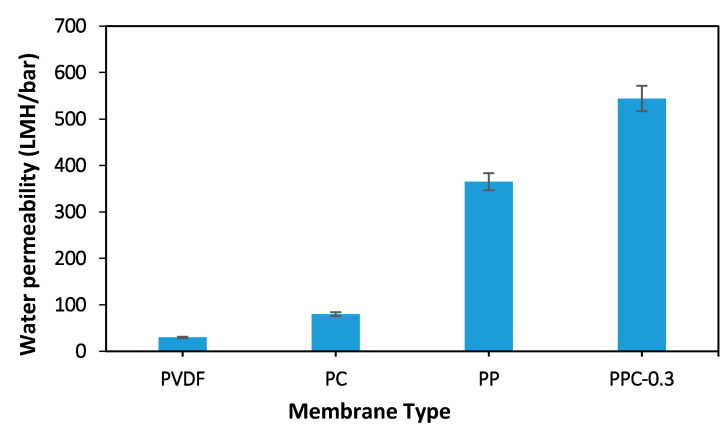
Water permeability for pristine and different types of membrane fabricated.

**Figure 10 membranes-12-00751-f010:**
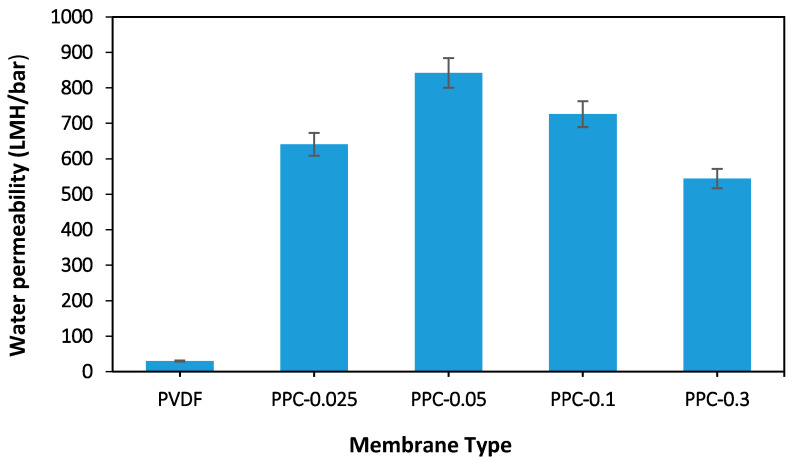
Water flux of the pristine and modified membrane.

**Figure 11 membranes-12-00751-f011:**
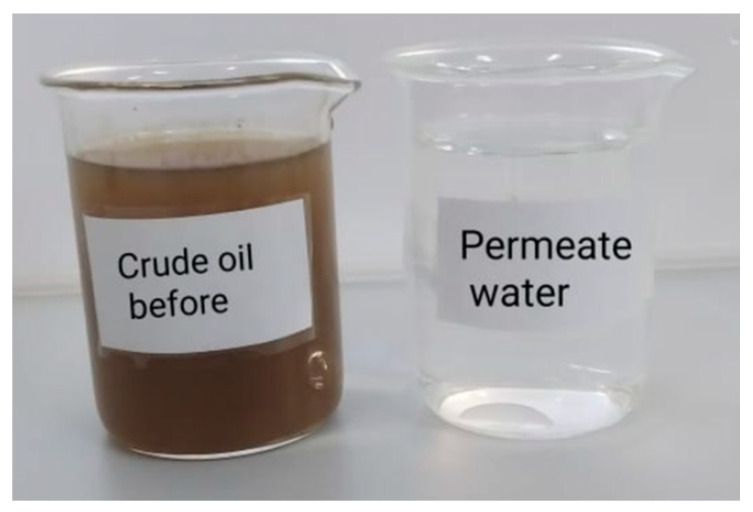
Feed and filtrate produced from a rejection of crude oil/water mixture.

**Figure 12 membranes-12-00751-f012:**
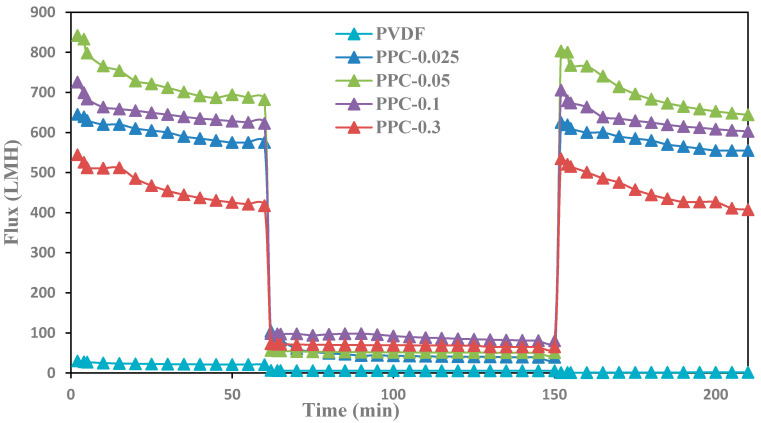
Fouling behavior and flux recovery for pristine and modified PVDF membrane: first one-hour for permeation with pure water, then rejection with crude oil, and finally, cleaning the membrane with 0.1M NaOH and one-hour permeation of pure water.

**Figure 13 membranes-12-00751-f013:**
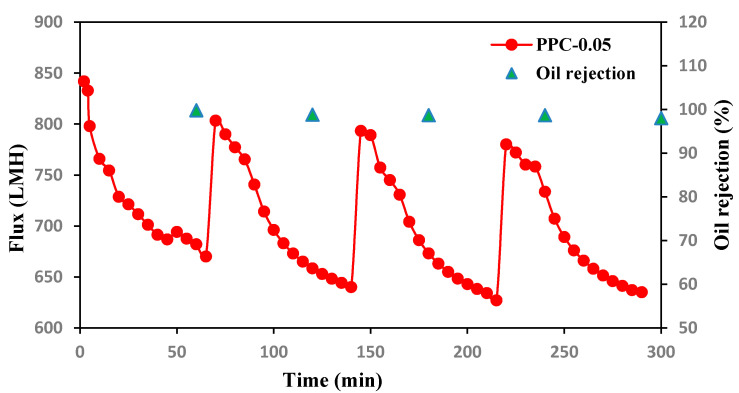
Oil Flux and rejection recycles for PPC-0.05 membrane versus time in Four filtration cycles after caustic soda cleaning.

**Figure 14 membranes-12-00751-f014:**
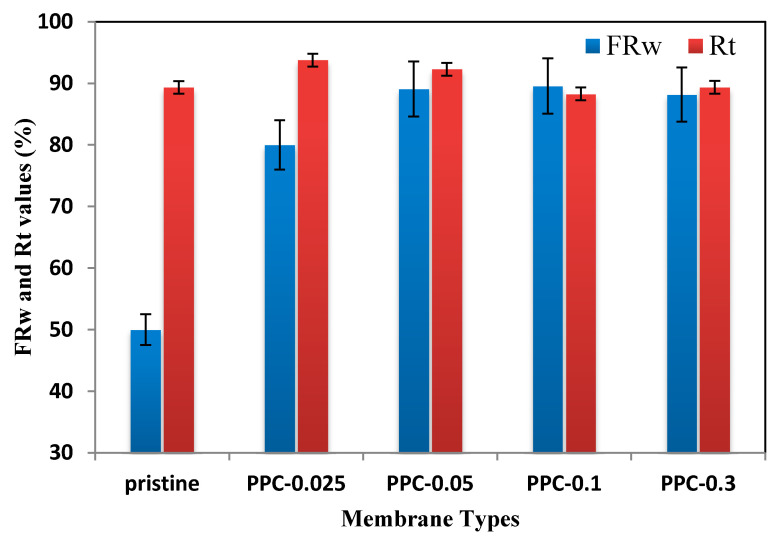
Total fouling ratio and Flux recovery for pure PVDF and resultant membranes.

**Table 1 membranes-12-00751-t001:** Compositions of different fabricated membranes.

Membrane Type	MWCNT (%)	PVDF (%)	Pyrrole (%)
**PVDF**	0	11.67	0
**PP**	0	11.67	3.67
**PPC-0.025**	0.025	11.67	3.67
**PPC-0.05**	0.05	11.67	3.67
**PPC-0.1**	0.1	11.67	3.67
**PPC-0.3**	0.3	11.67	3.67
**PC**	0.3	11.67	0

**Table 2 membranes-12-00751-t002:** Crude oil properties.

Crude Oil Properties	Droplet Size (nm)	Viscosity (cP)	pH	Zeta Potential (mV)
	400.1	0.8872	8.2	−15

**Table 3 membranes-12-00751-t003:** EDS elemental analysis for modified membranes.

Element	Weight Conc. (%)
C	56.00
N	4.20
O	8.80
F	31.00

**Table 4 membranes-12-00751-t004:** Membranes’ top layer pore size and porosity.

Membrane Type	Porosity (%)
**PVDF**	65.6 ± 2.8
**PP**	71.8 ± 3.0
**PPC-0.025**	81.6 ± 2.9
**PPC-0.05**	85.2 ± 2.0
**PPC-0.1**	89.9 ± 2.5
**PPC-0.3**	91.8± 2.0

**Table 5 membranes-12-00751-t005:** BET Pore size and Pore volume Studies.

Membrane Type	Pore Size (nm)	Pore Volume (cm^3^·g^−1^)
PVDF	3.05	0.00969
PPC-0.025	7.6	0.09389
PPC-0.05	8.25	0.01235
PPC-0.1	10.4	0.05991
PPC-0.3	9.2	0.01313

**Table 6 membranes-12-00751-t006:** Crude oil rejection by permeation cell.

Membrane	Crude Oil Rejection (%)
**PVDF**	90
**PPC-0.025**	99.5
**PPC-0.05**	99.8
**PPC-0.1**	99.9
**PPC-0.3**	99.9

**Table 7 membranes-12-00751-t007:** Comparison of different types of modified PVDF membrane used for oil removal with the current study.

Modified Membrane	Oil Rejection (%)	Flux Performance (L/m^2^ h)	Ref.
PVDF/PC	97.8%	22.11	[72]
PVDF/SiO_2_	98.99%	93.86	[73]
PVDF/Al_2_O_3_/PVP/sodium hexaphosphate	93.55%	138.53	[74]
PVDF/SiO_2_/PVP	94.5%	198	[75]
PVDF/MWCNT	99.89%	683.173	[13]
PVDF/PPy/MWCNT	99.9%	850	Current study

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
