# Peer review of "Fabrication of a Novel (PVDF/MWCNT/Polypyrrole) Antifouling High Flux Ultrafiltration Membrane for Crude Oil Wastewater Treatment"

_membranes, 2022, doi:10.3390/membranes12080751_

Round 1
Reviewer 1 Report
This manuscript reports the preparation of a composite UF membrane for crude oil wastewater treatment. It is demonstrated that the rejection rate of the MWCNT/ PPy /PVDF modified membrane can reach 99.9%. Some concerns and suggestions are specified as follows:
1. Most references cited in this manuscript, particularly on the nanocomposite membranes section, are too old to convincingly illustrate the latest research progress. It is suggested to review the most recent progress in the related fileds.
2. It is mentioned that the flux of the modified membrane decreases greatly during filtration, which requires shaking with alkaline solution for a long time to recover flux. Although PPy has good acid and alkali resistance, it is suggested to investigate the stable oil removal ability of the modified film after long-term soaking in alkaline solution.
3. I don’t think the pore size results measured by the gravimetric method is accurate or convincing. More standardized methods like MWCO measurement should be employed to determine UF membrane pore size.
4. What is the solution pH for the zeta potential measurement? Did the author conduct parallel sample measurements? Because the zeta potential varied significantly in different pH conditions.
5. It seems the discussion of “as shown previously in Fig.6 (+15.8 mV )” in line 514 referred to the wrong figure.
6. For Fig. 11, the antifouling experiments should be performed for at least three cycles to check is stability.
Reviewer 2 Report
Manuscript of Banan Hudaib et al. dedicated to the topical issue of obtaining water from systems containing hydrocarbons and solids, which are toxic to humans. For this, it is proposed to use membranes based on PVDF/MWCNT/PolyPyrrole; DMF is used as a solvent. Water acts as a precipitant. In this case, I recommend that the authors explain the choice of solvent and precipitant, which, as is well known, have a huge impact on the formed structure and morphology of the membrane.
The maximum content of MWCNT in the system is 0.3%.
The theoretical part of the manuscript, in my opinion, can be expanded and deepened into problems. The methodic part of the manuscript also requires more detailed disclosure. For example, the question arises why the authors do not determine the density of the polymer in the work, but use the literature data. How is the density of a composite material taken into account when calculating porosity?
The first thing the authors should pay attention to is the list of references and references in the text. At the moment they are not ordered and require appropriate actions.
Line 68. "COD" - you need to decipher the abbreviation.
2.1 materials. Authors should indicate more detailed characteristics of the materials used.
2.3 Oil preparation:. Maybe it's better for the authors to use "emulsion preparation" here?!
Lines 178-180. It needs to be checked.
Fig.4. Zeta potential for modified membranes. - I recommend to delete! This information is already presented in full in the text.
Fig.6. You need to add a scale bar to the photo and increase the font size.
Line 302. "Asymmetrical finger-like projections with large cavities and voids are well demonstrated in MWCNT/PPy/PVDF modified membranes." - the expression needs to be corrected.
3.5 mechanical properties of PVDF/MWCNT/PPy membranes. This section needs to be fixed!
Lines 446, 447. "PPC-0.1 showed a slight decrease to 726 LMH/bar, while for PPC-0.3, permeability decreased to 544 LMH/bar." - Why?
Lines 465 466. Please see Fig.9. replace with "(Fig.9)"
Lines 512, 513. "As crude oil wastewater is negatively charged molecules (the measured zeta potential is -15 mV)" - were these values ​​measured in this work? If not, then you must specify the data source.
In the final part of the manuscript, a comparison of the obtained data with membranes based on other polymers described in the literature is missing. I recommend authors to add relevant information.
Reviewer 3 Report
The manuscript ‘Fabrication of a Novel (PVDF/MWCNT/PolyPyrrole) Antifouling High Flux Ultrafiltration Membrane For Crude Oil Wastewater Treatment’ refers to a very interesting issue involving the prepare of a new type of membranes, but it is necessary to preparing the revised version of this article and I have a few comments that the authors should consider in preparing the revised version.
In the Introduction section should be corrected:
- It’s necessary preparation of the introduction based on the current literature.
In the Materials and Methods section should be explained:
- Why these process parameters and composition are used during membrane preparation?
- Why this composition of model solution was used in experiments?
- How thickness of the gold layer the samples were sprayed prior to the SEM observation?
- Why for the determination of the permeate flux, the notation F? There should be JP - the permate flux. Please revise markings throughout the manuscript.
- What solutions were used in determining the contact angle and zeta potential?
In the Results and Disccusion section should be supplemented:
The results of the study of elemental composition using EDS.
The discussion on the management of the retentate after the process - hazardous waste!
The influence of process and cleaning conditions on the properties of membranes using the same methods that were used to characterize unused membranes.
The discussion about membrane stability during water and wastewater treatment. Are components do not rinse out of the membrane, thus contaminating the purified stream.
Reviewer 4 Report
Review of the article entitled: Fabrication of a Novel (PVDF/MWCNT/PolyPyrrole) Antifouling High Flux Ultrafiltration Membrane For Crude Oil Wastewater Treatment
Authors: Banan Hudaib, Rund Abu-Zurayk, Haneen Waleed, Abed Al Qader Ibrahim
In my opinion, the reviewed manuscript fits the scope of Membranes journal. The paper is interesting and well organized. The Authors presented the results in the concise form, and the hypotheses were mostly confirmed by proper explanations and discussion with the literature survey. However, I have some comments that can help improve the manuscript:
1. The manuscript should be carefully checked in terms of many editor errors.
2. The captions style and format on the axes in all figures should be the same. Besides, there are no units on some axes captions.
3. In the caption of Fig. 6 the Authors wrote that the SEM images were recorded at 10 and 50 µm, however, in the pictures there are 30 µm. Moreover, in the last image (right d) there is no value. Please clarify this.
4. What was the stability of the prepared membranes? Did the Authors conduct the leakage test of nanomaterials from the membrane matrix?
5. Did the Authors carry out the tensile strength and elongation tests after cleaning membranes? Was there any negative effect of caustic soda on the membrane stability?
Reviewer 5 Report
Banan Hudaib et al. Fabrication of a Novel (PVDF/MWCNT/PolyPyrrole) Antifouling High Flux Ultrafiltration Membrane for Crude Oil Wastewater Treatment. The topic was interesting and promising results Therefore, it deserves to be published in Membranes. However, there are several minor issues that need to be corrected before it can be accepted for publication. Questions are as follows:
A revision of the Introduction should clearly identify the knowledge gaps addressed by this work and how it could fundamentally advance the field. The author needs to be cited the newest articles like MWCN-based ultrafiltration membranes and also UF membranes for oil wastewater treatment. Here are a few references you may cite Polymers 14.9 (2022): 1750 https://doi.org/10.3390/polym14091750, Bulletin of Materials Science 43 (2020): 1-12. (https://doi.org/10.1007/s12034-020-2079-7), International Journal of Biological Macromolecules 199 (2022): 36-41. https://doi.org/10.1016/j.ijbiomac.2021.12.087, Environmental Technology & Innovation 21 (2021): 101322 (https://doi.org/10.1016/j.eti.2020.101322)
· The contact angles units were given wrongly making it degree in superscript in lines 229 and 232. And figure 3 there is no unit for contact angles
· In Figure 8&9. the y-axis water permeability spelling was wrong please make it correct.
· What is the Crude Oil rejection of PP and PC membrane
· Fig. 11. Fouling behavior graph PVDF flux crossed the borders. Is it that the flux was negative values?
· Does the sieving separation help to separate oil drops? Because the oil rejection of pristine PVDF membrane is around 90% but it has the lowest pore size of 8 nm? Explain?
Round 2
Reviewer 1 Report
The authors have addressed my concerns
Author Response
Thank you very much
Reviewer 2 Report
The authors made corrections, but a number of comments were not answered in the required volume; additional comments and questions are also given below.
Line 18. Remove extra comma.
2.1 materials. Authors should indicate more detailed characteristics of the materials used. Thanks for your comment, more details are added to the material section - Authors should indicate polymer molecular weight, whether MWCNTs have been further processed, etc.
The authors need to justify the choice of a hard precipitant - water.
Line 206. "permeability for crude oil solution" - Needs to be checked and corrected!
Line 216. Replace "PPY" with "PPy"
Table3. To correct!
Fig.3. I propose to replace "FTIR" with "FTIR spectra".
Lines 283-300. How can the authors comment on the fact that the negative charge on the membrane surface should also prevent the penetration of oil?!
3.5 mechanical properties of PVDF/MWCNT/PPy membranes. This section needs to be fixed! - Thanks for your comment. The section is fixed to (Mechanical properties of modified membranes) in the manuscript.- No errors corrected!
The manuscript is not ready for publication and requires additional work with it!
Author Response
Dear reviewer,
Thank you for your comments
Please find replies in the attached file

Reviewer 3 Report
Accept in present form
Author Response
Thank you very much